# Building Carbon Emission Scenario Prediction Using STIRPAT and GA-BP Neural Network Model

**Sensen Zhang** 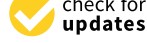**, Zhenggang Huo * and Chencheng Zhai**

College of Civil Science and Engineering, Yangzhou University, Yangzhou 225127, China;
ansel_98@163.com (S.Z.); mz120211019@yzu.edu.cn (C.Z.)
* Correspondence: mx120210609@stu.yzu.edu.cn

**Abstract:** As a major province of energy consumption and carbon emission, Jiangsu Province is also a major province of the construction industry, which is a key region and potential area for carbon emission reduction in China. The research and prediction of carbon emission in the construction industry is of great significance for the development of low-carbon policies in the construction industry of other cities. The purpose of this paper is to study the influencing factors of the whole life cycle carbon emissions of buildings in Jiangsu Province, and to predict the carbon emissions of buildings in Jiangsu Province based on the main influencing factors. This paper uses the energy balance sheet splitting method, STIRPAT model, gray correlation method and GA-BP neural network model to study and predict the carbon emissions of construction industry in Jiangsu Province. The research results show that the resident population, urbanization rate, steel production, average distance of road transportation, and labor productivity of construction enterprises have a catalytic effect on construction carbon emissions; GDP per capita and added value of tertiary industry have a suppressive effect; construction carbon emissions reached the historical peak in 2012; the prediction results show that the future construction carbon emissions in Jiangsu province generally show a decreasing trend. The research results of this paper provide a possibility to refine the study of construction carbon emission, and also provide a basis and guidance for subsequent research on construction carbon emission.

**Keywords:** building carbon emission; GA-BP neural network model; green building; building sustainability; scenario simulation

## 1. Introduction

According to a special report issued by the IPCC [1], the global climate temperature has increased by about 1.5 °C compared to the pre-industrial period. In order to cope with the rising global temperature, countries have formulated a series of emission reduction policies. As a major carbon emitter, China has been committed to energy conservation and emission reduction, fully demonstrating the responsibility and commitment of a significant country. In 2020, China committed at the UN General Assembly to strive to achieve peak carbon by 2030 and carbon neutrality by 2060.

According to the statistics of the Special Committee on Energy Consumption Statistics of China Building Energy Conservation Association [2], it is known that the whole process of carbon emission of the national building industry accounts for about 50% of the national carbon emission, and the construction industry has become a representative of high pollution and high energy consumption in China. The carbon emissions of China's buildings are shown in Figures 1 and 2. As a central province of energy consumption and carbon emission, Jiangsu Province is also a significant province of the construction industry, which is a crucial area and potential area for carbon emission reduction in China. Therefore, this paper intends to take the construction industry in Jiangsu Province as the research object and use the whole life cycle theory to divide the whole life cycle of construction into

the production stage of building materials, the transportation stage of building materials, the construction stage, the operation stage and the demolition stage of construction. The factors influencing the whole life cycle of construction in Jiangsu Province are studied, and the subsequent development trend is predicted.

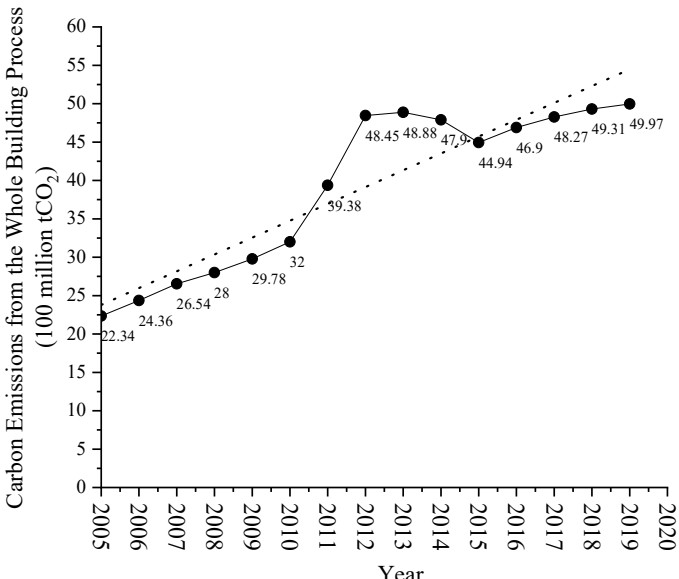

**Figure 1.** Carbon Emissions for the Whole Building Process 2005–2019.

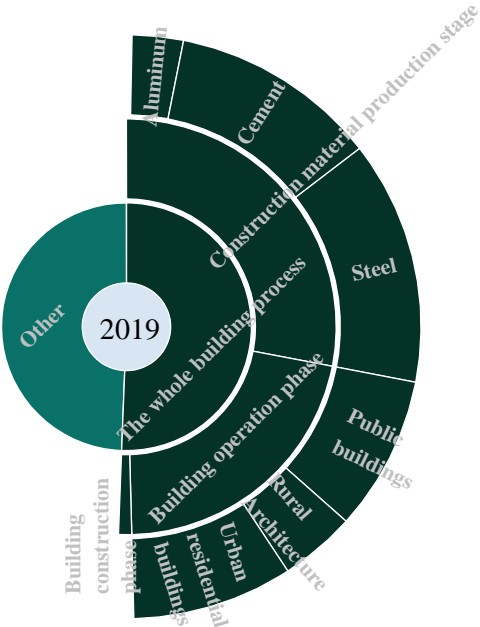

**Figure 2.** 2019 China Whole-Process Carbon Emissions for Construction.

## 2. Literature Review

Carbon emissions have become a hot topic in academia and an international issue. In order to accurately and effectively formulate national, regional, or industry carbon reduction policies, it is necessary to understand past and future carbon emission trends. Therefore, accounting for past carbon emissions and forecasting future carbon emissions has become particularly important. However, due to the caliber of energy statistics in China, there are difficulties in accounting for carbon emissions in the building sector, and the energy balance sheet splitting method is used in this study for accounting [3,4]. In addition, forming a reliable prediction model can provide a basis for policy or measure formulation for the country, regions, and industries, which facilitates the development

of special carbon emission reduction efforts. In the past decades, many scholars have conducted related studies, mainly using econometric as well as statistics-related principles and methods for simulation forecasting [5–9]. In recent years, more and more scholars have introduced the method of artificial neural networks into studying carbon emission prediction. Predictions using artificial neural network models trained on quarterly data from 1980 to 2015 in China found that population size was the most significant sensitive factor in China [10]. After ranking and differentiating the importance of influencing factors on CO2 emissions, carbon emission projections for China using dynamic nonlinear artificial neural networks found that the base case will peak in 2031 [11]. Factor decomposition based on the LMDI model and artificial neural network model predictions found carbon sinks, carbon capture and storage, and BECCS to be the preferred options for achieving carbon neutrality in China [12]. Predictive analysis based on IFWA-GRNN network models suggests that carbon intensity in 2030 is not attainable under the baseline scenario but is achievable under policy tightening and market allocation scenarios [13]. The prediction of the general regression neural network model combined with the urban development scenario found that economic development and the increase in construction industry standards may significantly impact future $CO_2$ emissions [14]. Combining the STIRPAT model and a GA-BP neural network model to forecast carbon emissions in Xinjiang, the results show that the low-carbon scenario is the first to peak, followed by the medium-carbon scenario, and the high-carbon scenario is difficult to peak before 2050 [15]. A BP neural network model was constructed to predict carbon emissions in Baoding, and the results found that: industrial production accounted for a larger share of carbon emissions, economic growth contributed significantly to the growth of carbon emissions, and energy intensity was the main negative factor [16].

The selection of drivers of carbon emissions as a precursor to carbon emission projections has also been the focus of carbon emission research. Energy consumption intensity, population growth, urbanization rate, and urban and rural GDP per capita were identified as the drivers of carbon emissions using the U-kaya equation [17]. The log-averaged Divisa index decomposition was used in the extended Kaya formula to include carbon emission factors, fossil fuel substitution, renewable energy penetration, carbon intensity, affluence, and population size as drivers [18]. The STIRPAT model was introduced into the carbon emission study to identify GDP per capita, total energy patents, economic accumulation, fossil fuel use, urbanization rate, and foreign direct investment as drivers. The IPAT-E model was used to identify and classify the drivers of carbon emissions as human impact, population, affluence, technology, and energy [19]. The extended IPAT model and scenario analysis were used to study carbon emission reduction pathways, including economic growth, population growth, energy intensity, and renewable energy share as drivers [20].

Green buildings, regarded as essential tools to combat climate change effectively, have been subject to extensive government incentives to promote their innovative development, and studies have found that supply-side policies can facilitate innovation and development of green building technologies as well as promote sustainable development in the construction industry [21]. In a comparison of green buildings with non-green buildings, it was found that carbon emissions from the operation and maintenance phases account for a larger share of the whole life cycle and that green buildings have slightly higher implied $CO_2$ emissions than non-green buildings, but their operational emissions are much lower [22]. Therefore, promoting the development of green building manufacturing can be an effective response to the deterioration of the climate environment, and studies have shown that "energy efficiency and "indoor environmental quality" are sustainability indicators in green building manufacturing [23], in practice, we can grasp these indicators to promote the development of green building industry better. An analysis based on the RBF-WINGS model found that industry scale and green financial support are the main influencing factors for green building development [24], which can provide a theoretical basis for the development of green buildings in China and a reference for government departments to make decisions. However, the high cost is still the biggest obstacle to

implementing green buildings in developing countries [25]. In the case of the Murabba Palace, a number of building interventions incorporating passive design were found: the use of double walls with double low-e glazing and the application of polystyrene insulation improved the thermal comfort of the building interior and contributed to the reduction of carbon emissions [26].

In summary, most of the studies on carbon emission projections are still conducted at the national level, there are fewer studies on regional or industry sectors, and there is a lack of accounting and projections of building carbon emissions for each province and city in China. In addition, most studies on carbon emission calculation and projection seldom consider the dynamic changes of electricity and thermal power carbon emission factors.

## 3. Research Methods and Models

### 3.1. Gray Correlation Method

The basic idea of the gray correlation degree method is to judge whether the connections are strong or not based on the similarity of the shapes of the set of sequential curves [27]. Since there are many influencing factors on carbon emission in each stage of the whole life cycle of a building, this paper applies this method aiming to select the main influencing factors in each stage of the whole life cycle of a building to construct the STIRPAT model.

The basic idea [28]: determine the evaluation object as well as the evaluation index; perform standardization to remove the scale; calculate the absolute difference between the evaluation object index series and the reference series; calculate the gray correlation coefficient using Equation (1); calculate the gray correlation degree using Equation (2); and perform evaluation analysis.

$$\zeta_i(k) = \frac{\displaystyle\min_i \min_k |x_0(k) - x_i(k)| + \rho \min_i \min_k |x_0(k) - x_i(k)|}{|x_0(k) - x_i(k)| + \rho \min_i \min_k |x_0(k) - x_i(k)|} \tag{1}$$

$$r_i = \sum_{k=1}^{K} w_k \zeta_i(k) \tag{2}$$

This paper takes the production stage of building materials as an example and uses the gray correlation method to select the main influencing factors in this stage, the same for the other stages. The consumption of construction materials and carbon emissions from the production of building materials in Jiangsu Province over the years are shown in Table 1.

**Table 1.** Consumption of construction materials and carbon emissions from building materials production in Jiangsu Province over the years.

| Year | Steel (Ton) | Cement (Ton) | Glass (Weight Box) | Aluminum (Ton) | Carbon Emissions (Ten Thousand Tons) |
|------|-------------|--------------|--------------------|----------------|--------------------------------------|
| 2005 | 17,176,219 | 72,850,624 | 11,755,245 | 2,135,532 | 12,640.5656 |
| 2006 | 19,365,741 | 82,523,173 | 5,807,113 | 722,611 | 10,412.6537 |
| 2007 | 24,896,823 | 91,657,074 | 6,090,033 | 829,837 | 12,292.2506 |
| 2008 | 30,475,791 | 118,258,785 | 10,392,076 | 780,865 | 14,892.7165 |
| 2009 | 35,532,626 | 152,498,685 | 16,587,859 | 1,079,470 | 18,644.9787 |
| 2010 | 40,618,866 | 157,705,493 | 17,325,388 | 1,301,559 | 20,465.4572 |
| 2011 | 47,989,650 | 182,777,750 | 14,408,094 | 1,738,724 | 24,388.8858 |
| 2012 | 141,117,794 | 754,308,672 | 18,685,165 | 7,466,010 | 89,863.4490 |
| 2013 | 130,393,503 | 316,346,453 | 20,494,290 | 17,162,732 | 83,597.1095 |
| 2014 | 79,550,112 | 245,611,224 | 177,321,086 | 4,397,407 | 41,305.1479 |

**Table 1.** *Cont.*

| Year | Steel (Ton) | Cement (Ton) | Glass (Weight Box) | Aluminum (Ton) | Carbon Emissions (Ten Thousand Tons) |
|------|-------------|--------------|--------------------|-----------------|---------------------------------------|
| 2015 | 86,746,631 | 271,234,936 | 22,840,682 | 4,136,138 | 42,787.4867 |
| 2016 | 90,703,898 | 245,411,043 | 26,027,869 | 5,243,690 | 44,554.2099 |
| 2017 | 90,562,639 | 241,763,822 | 20,947,231 | 4,293,637 | 42,153.9950 |
| 2018 | 93,709,279 | 239,359,027 | 25,063,438 | 4,493,855 | 43,079.6396 |
| 2019 | 100,465,171 | 257,666,274 | 26,811,036 | 4,107,839 | 44,638.2866 |

The gray correlation between steel, cement, glass, and aluminum and carbon emissions are 0.9588, 0.9373, 0.8524, and 0.9130, respectively. Therefore, steel production can be selected as the influencing factor of carbon emissions in the production stage of building materials.

### 3.2. STIRPAT Model

STIRPAT [29] is an extensible stochastic environmental impact assessment model that can assess the relationship between human factors and the environment [30]. In this paper, the STIRPAT model is applied to construct a relationship model between carbon emissions of buildings and their influencing factors as follows.

$$CE = a \times P^p \times C^c \times G^g \times T^t \times S^s \times R^r \times L^l \times e \tag{3}$$

Taking logarithms of both sides of the above equation simultaneously yields:

$$lnCE = lna + plnP + clnC + glnG + tlnT + slnS + rlnR + llnL + lne \tag{4}$$

where *CE* refers to carbon emissions from buildings; *P* refers to resident population; *C* refers to urbanization rate; *G* refers to GDP per capita; *T* refers to value-added of tertiary industry; *S* refers to steel production; *R* refers to an average distance of road transportation; L refers to labor productivity of construction enterprises; *p*, *c*, *g*, *t*, *s*, *r*, and *l* refer to the elasticity coefficients of each index; is a constant term, and is an error term.

### 3.3. GA-BP Neural Network Model

This paper uses a GA-BP neural network model to predict the development trend of building carbon emission in Jiangsu province, which is a multi-layer feed-forward neural network trained according to the error back propagation algorithm after optimizing weights and thresholds by a genetic algorithm, and the initial weights and thresholds of BP neural network are optimized by a genetic algorithm to reduce the model error and improve the accuracy of prediction. The flow is shown in Figure 3.

#### 3.3.1. BP Neural Network

The working process of the BP neural network is roughly divided into two subprocesses: forward transmission of the working signal and reverse transmission of the error signal. In the BP neural network constructed in this paper, each sample has seven inputs and one output, and the number of neurons in the hidden layer is determined as fifteen according to Equation (5). Robert Hecht-Nielsen found that a continuum in any closed interval can be approximated by a BP neural network with a hidden layer [31]. Therefore, in this paper, a three-layer BP network is chosen, and its basic structure is shown in Figures 4 and 5.

$$S = 2 \times N + 1 \tag{5}$$

where *N* denotes the number of neurons in the input layer; *S* is the number of neurons in the hidden layer.

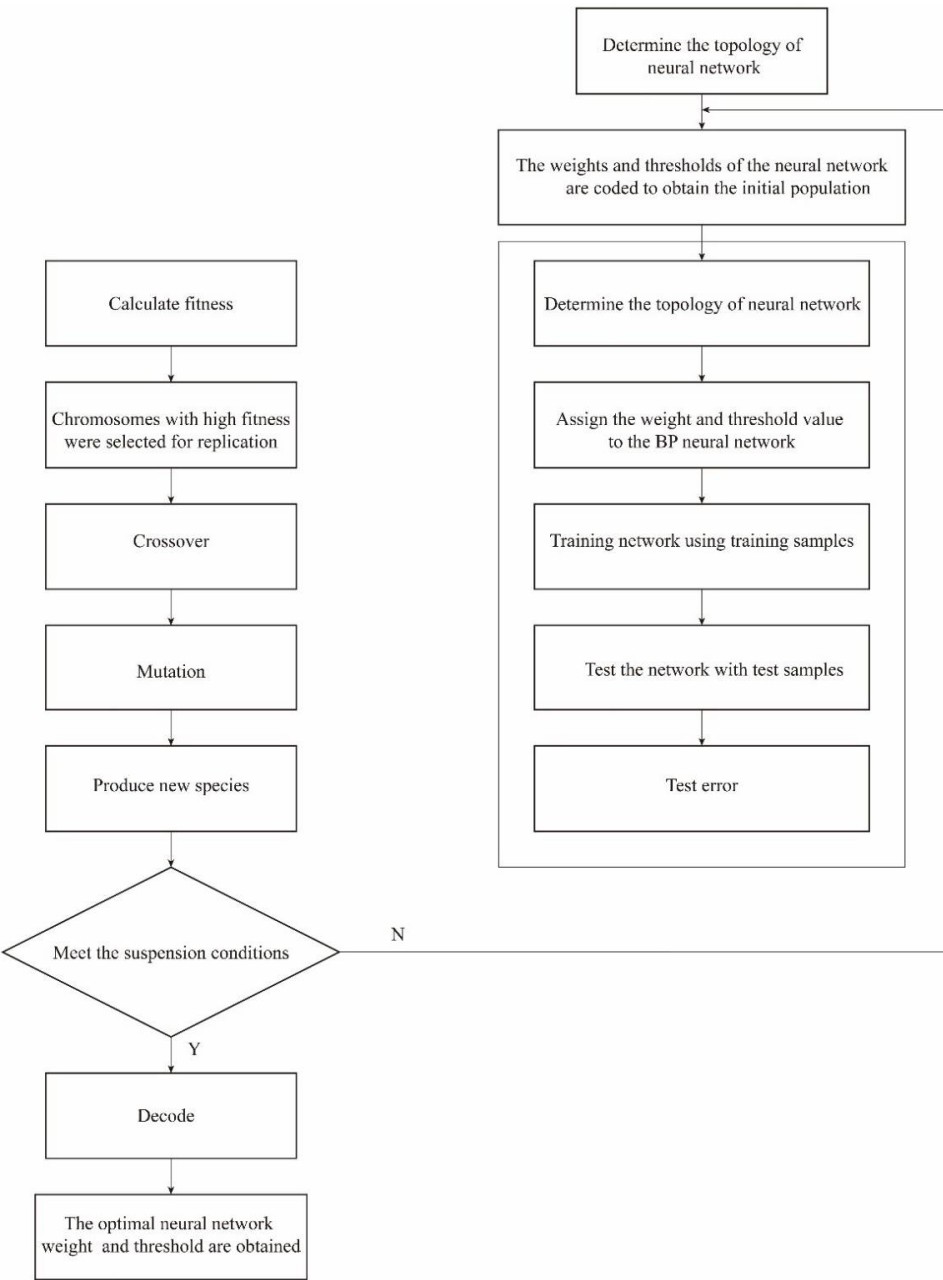

**Figure 3.** Basic process of GA-BP neural network algorithm.

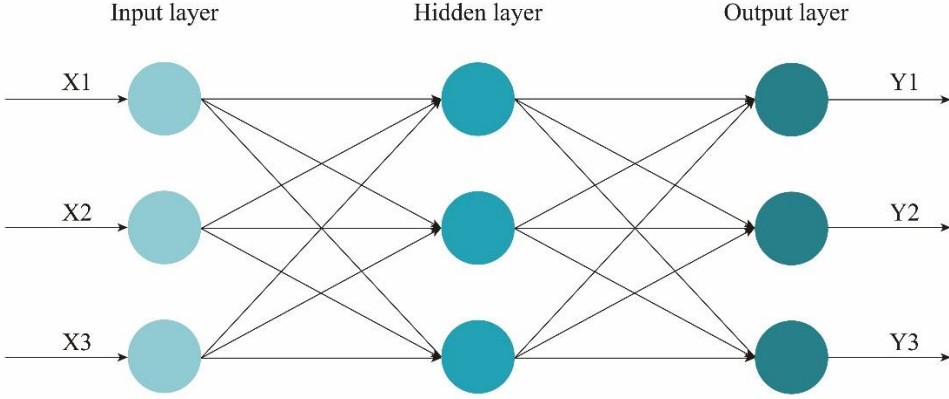

**Figure 4.** BP neural network basic structure.

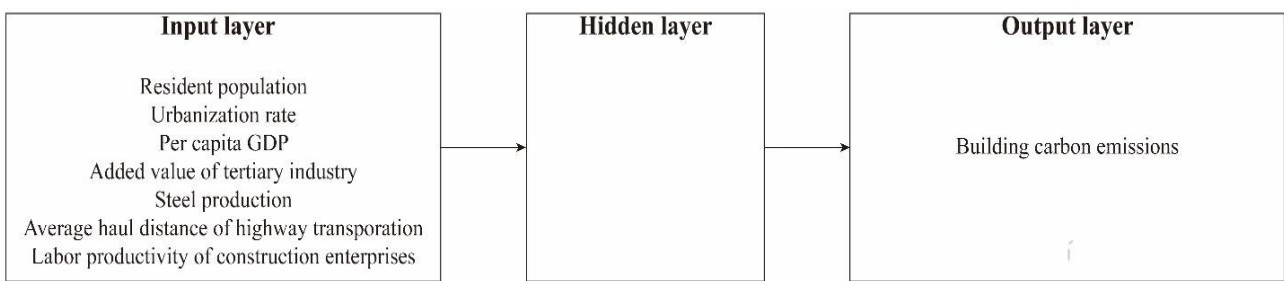

**Figure 5.** BP neural network structure for building carbon emission prediction.

The basic computational steps of the BP neural network are as follows.

Step 1. Network initialization. In this study, the number of nodes in the input layer is seven, the number of nodes in the hidden layer is fifteen, and the number of nodes in the output layer is one. The weight of the input layer to the hidden layer is $w_{ij}$, the weight of the hidden layer is $w_{jk}$, the threshold of the input layer to the hidden layer is $b_j$, and the threshold of the hidden layer to the output layer is $b_k$. The learning rate is η, the activation function is *f(x)*, and the activation function in this study is chosen as the Sigmoid function shown in Equation (6).

$$f(x) = \frac{1}{1 + e^{-x}} \tag{6}$$

Step 2. Calculate the output value $H_j$ of the hidden layer.

$$H_j = f\left(\sum_{i=1}^{7} w_{ij} x_i + b_j\right) \tag{7}$$

Step 3. Compute the output value $O_k$ of the output layer.

$$O_k = \sum_{j=1}^{15} H_j w_{jk} + b_k \tag{8}$$

Step 4. Calculate the error between the output value $O_k$ and the desired output value $EO_k$.

$$E = \frac{1}{2}(EO_k - O_k)^2 \tag{9}$$

Step 5. The weights are updated according to the error back propagation between the output value and the desired output value, using the gradient descent method.

$$\frac{\partial E}{\partial w_{jk}} = (EO_k - O_k)\left(-\frac{\partial O_k}{\partial w_{jk}}\right) = (EO_k - O_k)(-H_j) \tag{10}$$

$$\frac{\partial E}{\partial w_{ij}} = \frac{\partial E}{\partial H_j} \cdot \frac{\partial H_j}{\partial w_{ij}} \tag{11}$$

$$\frac{\partial E}{\partial H_j} = (EO_k - O_k)\left(-\frac{\partial O_k}{\partial H_j}\right) = -w_{jk}(EO_k - O_k) \tag{12}$$

$$\frac{\partial H_j}{\partial w_{ij}} = \frac{\partial f\left(\sum_{i=1}^{7} w_{ij} x_i + b_j\right)}{\partial w_{ij}} = H_j(1 - H_j) x_i \tag{13}$$

$$\begin{cases} w_{ij} = w_{ij} + \eta H_j(EO_k - O_k) \\ w_{ij} = w_{ij} + \eta H_j(1 - H_j) x_i w_{ij}(EO_k - O_k) \end{cases} \tag{14}$$

Step 6. The error between the output value and the desired output value is back propagated and gradient descent is used to update the threshold value.

$$\frac{\partial E}{\partial b_k} = (EO_k - O_k)\left(-\frac{\partial O_k}{\partial b_k}\right) = -(EO_k - O_k) \tag{15}$$

$$\frac{\partial E}{\partial b_j} = \frac{\partial E}{\partial H_j} \cdot \frac{\partial H_j}{\partial b_j} \tag{16}$$

$$\frac{\partial H_j}{\partial b_j} = \frac{\partial f(\sum_{i=1}^{7} w_{ij} x_i + b_j)}{\partial b_j} = H_j(1 - H_j) \tag{17}$$

$$\frac{\partial E}{\partial H_j} = (EO_k - O_k)\left(-\frac{\partial O_k}{\partial H_j}\right) = -w_{jk}(EO_k - O_k) \tag{18}$$

$$\begin{cases} b_k = b_k + \eta(EO_k - O_k) \\ b_j = b_j + \eta H_j(1 - H_j) w_{jk}(EO_k - O_k) \end{cases} \tag{19}$$

Step 7. Determine whether the algorithm iteration is completed.

### 3.3.2. Genetic Algorithm

Step 1. The connection with the BP neural network is established through the fitness function, and in this study, the fitness function is associated with the error in the neural network model, as shown in Equation (20).

$$fitness = \frac{1}{E} = \frac{2}{(EO_k - O_k)^2} \tag{20}$$

Step 2. In this study, chromosome selection and duplication were performed using the roulette wheel method, while retaining the elite chromosomes in the sire for inheritance. The basic calculation of roulette is shown below.

$$p(x_i) = \frac{f(x_i)}{\sum_{j=1}^{N} f_i} \tag{21}$$

$$q(x_i) = \sum_{j=1}^{i} p(x_j) \tag{22}$$

where $p(xi)$ denotes the probability of each individual being selected; and $q(xi)$ denotes the cumulative probability of each component.

Step 3. Select the parent chromosome for crossover to produce new offspring.
Step 4. Mutate the chromosomes of the offspring.
Step 5. Repeat Step 2, Step 3, Step 4 until a new population is generated.

### 3.3.3. Genetic Algorithm to Optimize BP Neural Network

Step 1. Determine the structure of BP neural network.
Step 2. Optimization of weights and thresholds using genetic algorithm.
Step 3. The optimized BP neural network is trained and predicted.

### 3.4. Ridge Regression

Ridge regression [32] was first proposed by Hoerl et al. in 1970 as an improved least squares estimation that obtains regression coefficients at the cost of losing some information and reducing accuracy. Unlike general linear regression, the unbiased estimation of ridge regression tends to shrink some of the coefficients toward zero, which helps to alleviate the problems of multiple co-linearity and overfitting.

The prediction model of building carbon emissions in Jiangsu Province is built according to this process, as shown in Figure 6.

| **Method** | **Content** |
|---|---|
| Energy balance sheet splitting method | STEP 1<br>Calculating the whole life cycle carbon emissions of buildings in Jiangsu Province from 2005 to 2019 |
| Literature review | STEP 2<br>Analyzing the fatcors affecting the carbon emissions of buildings in Jiangsu Province from 2005 to 2019 |
| Gray correlation method | STEP 3<br>Selecting the main influencing factors at each stage of the whole life cycle of budings |
| STIRPAT model<br>Ridge regression | STEP 4<br>Constructing STIRPAT model and analyzing the relationship between each influencing factor and carbon emissions |
| GA-BP neural network model | STEP 5<br>Forcasting the carbon emissions of construciton in Jiangsu Province from 2020 to 2024 |

**Figure 6.** The flow chart of proposed methodology.

## 4. Model Construction and Data Analysis

All raw data used in this paper are from China Statistical Yearbook, Jiangsu Statistical Yearbook, China Construction Statistical Yearbook, and China Energy Statistical Yearbook from 2005–2019.

In this paper, the energy balance sheet splitting method [2,3,33,34] is used to calculate the whole life cycle of building carbon emissions. Among them, the energy carbon emission factor is calculated using Equation (23) [35]; the electricity carbon emission factor is calculated using Equation (24); and the thermal carbon emission factor is calculated using Equation (25).

$$f = 44/12 \times J \times C \times O \tag{23}$$

$$f_e = CE_t/(E_t + E_r) \tag{24}$$

$$f_h = CE_h/P_h \tag{25}$$

where, $f$ refers to energy carbon emission factor; $J$ refers to average low-level heat generation; $C$ refers to carbon content per unit calorific value; $O$ refers to carbon oxidation rate;

$f_e$ refers to electricity carbon emission factor; *CEt* refers to carbon emission from thermal power generation; *Et* refers to thermal power generation; $E_r$ refers to renewable energy generation. $f_h$ refers to thermal carbon emission factor; $CE_h$ refers to carbon emission from various types of heating energy; and *Ph* refers to total heat production.

The carbon emission factors of electricity and thermal power in Jiangsu Province are shown in Tables 2 and 3 respectively. The carbon emissions from the production of construction materials, transportation of construction materials, building operation and building construction processes in Jiangsu Province are shown in Tables 4–7 respectively. The raw data of building carbon emissions and their influencing factors in Jiangsu Province are shown in Table 8.

**Table 2.** Carbon Emission Factor for Electricity in Jiangsu Province.

| Year | 2005 | 2006 | 2007 | 2008 | 2009 |
|---|---|---|---|---|---|
| Power carbon emission factor (kgCO$_2$/kWh) | 0.8879 | 0.8517 | 0.7798 | 0.7630 | 0.7394 |
| Year | 2010 | 2011 | 2012 | 2013 | 2014 |
| Power carbon emission factor (kgCO$_2$/kWh) | 0.7082 | 0.7475 | 0.7430 | 0.7330 | 0.6972 |
| Year | 2015 | 2016 | 2017 | 2018 | 2019 |
| Power carbon emission factor (kgCO2/kWh) | 0.7007 | 0.6869 | 0.6841 | 0.6556 | 0.6455 |

**Table 3.** Thermal Carbon Emission Factor for Jiangsu Province.

| Year | 2005 | 2006 | 2007 | 2008 | 2009 |
|---|---|---|---|---|---|
| Thermal carbon emission factor (kg/kg) | 3.1312 | 3.2555 | 3.0945 | 3.1505 | 3.0677 |
| Year | 2010 | 2011 | 2012 | 2013 | 2014 |
| Thermal carbon emission factor (kg/kg) | 2.8153 | 2.9660 | 3.0769 | 3.0159 | 3.0128 |
| Year | 2015 | 2016 | 2017 | 2018 | 2019 |
| Thermal carbon emission factor (kg/kg) | 3.0690 | 3.0149 | 3.0935 | 3.6314 | 3.5692 |

**Table 4.** Carbon emissions from building materials production in Jiangsu Province.

| Year | 2005 | 2006 | 2007 | 2008 | 2009 |
|---|---|---|---|---|---|
| Carbon emissions (Ten Thousand Tons) | 12,640.5656 | 10,412.6537 | 12,292.2506 | 14,892.7165 | 18,644.9787 |
| Year | 2010 | 2011 | 2012 | 2013 | 2014 |
| Carbon emissions (Ten Thousand Tons) | 20,465.4572 | 24,388.8858 | 89,863.4490 | 83,597.1095 | 41,305.1479 |
| Year | 2015 | 2016 | 2017 | 2018 | 2019 |
| Carbon emissions (Ten Thousand Tons) | 42,787.4867 | 44,554.2099 | 42,153.9950 | 43,079.6396 | 44,638.2686 |

**Table 5.** Carbon emission of building materials transportation in Jiangsu Province.

| Year | 2005 | 2006 | 2007 | 2008 | 2009 |
|---|---|---|---|---|---|
| Carbon emissions (Ten Thousand Tons) | 113.9471 | 136.4962 | 158.0850 | 197.8498 | 320.4802 |
| Year | 2010 | 2011 | 2012 | 2013 | 2014 |
| Carbon emissions (Ten Thousand Tons) | 345.2539 | 423.8154 | 1664.8701 | 1353.2409 | 978.1925 |
| Year | 2015 | 2016 | 2017 | 2018 | 2019 |
| Carbon emissions (Ten Thousand Tons) | 1013.6430 | 942.5937 | 961.7008 | 941.2930 | 1070.3047 |

**Table 6.** Carbon emission from building operation in Jiangsu Province.

| Year | 2005 | 2006 | 2007 | 2008 | 2009 |
|---|---|---|---|---|---|
| Carbon emissions (Ten Thousand Tons) | 4127.0743 | 4359.5150 | 4511.4667 | 5160.9920 | 5570.8610 |
| Year | 2010 | 2011 | 2012 | 2013 | 2014 |
| Carbon emissions (Ten Thousand Tons) | 6139.5220 | 6872.6039 | 7680.1223 | 8694.3287 | 8019.8521 |
| Year | 2015 | 2016 | 2017 | 2018 | 2019 |
| Carbon emissions (Ten Thousand Tons) | 9013.4497 | 9462.6038 | 10,492.9558 | 11,328.2717 | 11,616.3580 |

**Table 7.** Carbon emission during construction in Jiangsu Province.

| Year | 2005 | 2006 | 2007 | 2008 | 2009 |
|---|---|---|---|---|---|
| Carbon emissions * (Ten Thousand Tons) | 449.1982 | 473.4452 | 483.9768 | 499.4249 | 530.2938 |
| Year | 2010 | 2011 | 2012 | 2013 | 2014 |
| Carbon emissions (Ten Thousand Tons) | 648.4856 | 771.1489 | 852.3013 | 944.6358 | 971.8402 |
| Year | 2015 | 2016 | 2017 | 2018 | 2019 |
| Carbon emissions (Ten Thousand Tons) | 901.2428 | 828.8998 | 855.2922 | 901.7684 | 953.4604 |

* Carbon emissions in the construction stage include carbon emissions from construction and demolition.

**Table 8.** Raw data of influencing factors and building carbon emissions in Jiangsu Province.

| Year | Resident Population (Ten Thousand People) | Urbanization Rate (%) | Per Capita GDP (Yuan) | Added Value of Tertiary Industry (Hundred Million Yuan) | Steel Production (Ton) | Average Distance of Highway Transportation (km) | Labor Productivity of Construction Enterprises (Yuan/Person) | Building Carbon Emissions (Ten Thousand Tons) |
|---|---|---|---|---|---|---|---|---|
| 2005 | 7588.24 | 50.5 | 23,984 | 1300.10 | 17,176,219 | 60.18 | 127,657 | 17,330.79 |
| 2006 | 7655.66 | 51.9 | 27,868 | 1284.32 | 19,365,741 | 64.29 | 142,846 | 15,382.11 |
| 2007 | 7723.13 | 53.2 | 33,798 | 1993.99 | 24,896,823 | 65.51 | 160,387 | 17,445.78 |
| 2008 | 7762.48 | 54.3 | 39,967 | 2130.80 | 30,475,791 | 65.60 | 174,742 | 20,750.98 |
| 2009 | 7810.27 | 55.6 | 44,272 | 1754.71 | 35,532,626 | 93.38 | 189,932 | 25,066.61 |
| 2010 | 7869.34 | 60.6 | 52,787 | 3460.14 | 40,618,866 | 93.04 | 207,116 | 27,598.72 |
| 2011 | 8022.99 | 62.0 | 61,464 | 3578.19 | 47,989,650 | 93.41 | 249,338 | 32,456.45 |
| 2012 | 8119.81 | 63.0 | 66,533 | 2610.53 | 141,117,794 | 94.50 | 262,834 | 100,060.74 |
| 2013 | 8192.44 | 64.4 | 72,768 | 3442.75 | 130,393,503 | 172.64 | 282,532 | 94,589.31 |
| 2014 | 8281.09 | 65.7 | 78,711 | 3421.77 | 79,550,112 | 172.87 | 296,918 | 51,275.03 |
| 2015 | 8315.11 | 67.5 | 85,871 | 3757.42 | 86,746,631 | 182.88 | 297,437 | 53,715.82 |
| 2016 | 8381.47 | 68.9 | 92,658 | 4337.88 | 90,703,898 | 182.68 | 304,925 | 55,788.31 |
| 2017 | 8423.50 | 70.2 | 102,202 | 4430.92 | 90,562,639 | 184.45 | 312,383 | 54,463.94 |
| 2018 | 8446.19 | 71.2 | 110,508 | 4235.98 | 93,709,279 | 182.72 | 335,803 | 56,250.97 |
| 2019 | 8469.09 | 72.5 | 116,650 | 3915.58 | 100,465,171 | 196.55 | 363,015 | 58,278.39 |

### 4.1. Multicollinearity Analysis

Tables 9 and 10 were obtained after multiple regression analysis using SPSS software. *lnP, lnC, lnG, lnT, lnR, and lnL* in Table 9 had tolerances less than 0.1 and VIF values greater than 10; the maximum value of the conditional index in Table 10 was 8050.731, which indicated the existence of multicollinearity among the independent variables.

### 4.2. Ridge Regression Analysis

In this paper, SPSSPRO software was used to perform ridge regression analysis, and the results of the analysis are shown in Figure 7 and Table 11. The *lnP, lnC, lnG, lnT, lnS, lnR,* and *lnL* regression model significance *p* value is 0.001, which indicates significance at a high level and rejects the original hypothesis, indicating that there is a regression relationship between the independent and dependent variables. Meanwhile, the goodness of fit $R^2$ of the model is 0.944, and the model performs better.

**Table 9.** Coefficient a.

| Model | Non-Standardized Coefficient | | | t | Significance | Collinearity Statistics | |
|---|---|---|---|---|---|---|---|
| | B | Standard Error | Standardization Coefficient | | | Tolerance | VIF |
| Constant | −5.646 | 31.242 | | −0.181 | 0.862 | | |
| *lnP* | −0.412 | 3.935 | −0.026 | −0.105 | 0.920 | 0.013 | 79.689 |
| *lnC* | 2.826 | 1.550 | 0.562 | 1.823 | 0.111 | 0.008 | 121.781 |
| *lnG* | −1.027 | 0.447 | −0.843 | −2.296 | 0.055 | 0.006 | 172.917 |
| *lnT* | −0.092 | 0.135 | −0.064 | −0.682 | 0.517 | 0.088 | 11.307 |
| *lnS* | 1.110 | 0.074 | 1.261 | 14.895 | 0.000 | 0.109 | 9.195 |
| *lnR* | 0.089 | 0.136 | 0.068 | 0.655 | 0.534 | 0.073 | 13.672 |
| *lnL* | 0.006 | 0.590 | 0.003 | 0.009 | 0.993 | 0.008 | 130.036 |

**Table 10.** Collinearity diagnosis.

| Dimension | Characteristic Value | Condition Index | Variance Ratio | | | | | | | |
|---|---|---|---|---|---|---|---|---|---|---|
| | | | Constant | *lnP* | *lnC* | *lnG* | *lnT* | *lnS* | *lnR* | *lnL* |
| 1 | 7.993 | 1.000 | 0.00 | 0.00 | 0.00 | 0.00 | 0.00 | 0.00 | 0.00 | 0.00 |
| 2 | 0.006 | 36.337 | 0.00 | 0.00 | 0.00 | 0.00 | 0.00 | 0.00 | 0.07 | 0.00 |
| 3 | 0.001 | 122.863 | 0.00 | 0.00 | 0.00 | 0.00 | 0.30 | 0.00 | 0.24 | 0.00 |
| 4 | 0.000 | 168.490 | 0.00 | 0.00 | 0.00 | 0.00 | 0.05 | 0.42 | 0.10 | 0.00 |
| 5 | $7.258 \times 10^{-5}$ | 331.866 | 0.00 | 0.00 | 0.01 | 0.08 | 0.42 | 0.21 | 0.28 | 0.00 |
| 6 | $7.421 \times 10^{-6}$ | 1037.830 | 0.00 | 0.00 | 0.63 | 0.08 | 0.04 | 0.09 | 0.04 | 0.16 |
| 7 | $3.574 \times 10^{-6}$ | 1495.421 | 0.00 | 0.00 | 0.13 | 0.79 | 0.01 | 0.27 | 0.03 | 0.84 |
| 8 | $1.233 \times 10^{-7}$ | 8050.731 | 0.99 | 1.00 | 0.22 | 0.04 | 0.19 | 0.01 | 0.25 | 0.00 |

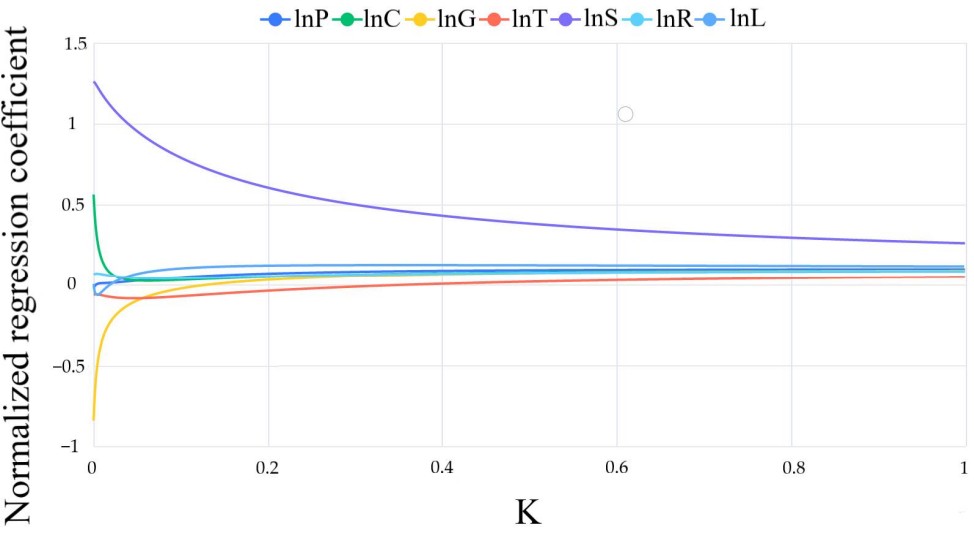

**Figure 7.** Ridge trace map.

**Table 11.** Ridge regression analysis results.

| K = 0.127 | Non-Standardized Coefficient | | Standardization Coefficient | t | *p* | R2 | Adjusted R2 | F |
|---|---|---|---|---|---|---|---|---|
| | B | Standard Error | Beta | | | | | |
| constant | −11.544 | 9.752 | - | −1.184 | 0.275 | | | |
| lnP | 0.877 | 1.146 | 0.056 | 0.765 | 0.469 | | | |
| lnC | 0.18 | 0.309 | 0.036 | 0.581 | 0.580 | | | |
| lnG | −0.004 | 0.073 | −0.004 | −0.061 | 0.953 | 0.944 | 0.889 | 17.001 (0.001 ***[1]) |
| lnT | −0.091 | 0.161 | −0.063 | −0.561 | 0.592 | | | |
| lnS | 0.639 | 0.1 | 0.726 | 6.402 | 0.000 ***[1] | | | |
| lnR | 0.057 | 0.145 | 0.044 | 0.396 | 0.704 | | | |
| lnL | 0.207 | 0.115 | 0.112 | 1.801 | 0.115 | | | |

[1] *** represent the significance level of 1%.

The resulting model equation is as follows:

$$lnCE = -11.554 + 0.877lnP + 0.18lnC - 0.004lnG - 0.091lnT + 0.639lnS \\ + 0.057lnR + 0.207lnL \tag{26}$$

$$CE = (9.5976e - 06) \times P^{0.877} \times C^{0.18} \times G^{-0.004} \times T^{-0.091} \times S^{0.639} \times R^{0.057} \\ \times L^{0.207} \tag{27}$$

## 5. Carbon Emission Scenario Setting for Buildings in Jiangsu Province

In this paper, the scenario model is set to low-carbon, baseline, and high-carbon, and the rate of change of each factor is set based on the overall trend and annual average rate of change of the factor in the past ten years, with reference to the relevant policies and plans promulgated by the state and Jiangsu Province. The scenario settings for carbon emission prediction in Jiangsu Province are shown in Table 12.

**Table 12.** Scenario setting of building carbon emission prediction in Jiangsu Province.

| Scenario Pattern | Rate of Change (%) | | | | | | |
|---|---|---|---|---|---|---|---|
| | Population Size | Urbanization Rate | Per Capita GDP | Added-Value of Tertiary Industry | Steel Production | Average Haul Distance of Highway Transportation | Labor Productivity of Construction Enterprises |
| Low carbon | 0.70 | 1.25 | 10.5 | 9.5 | 18 | 9.5 | 6.5 |
| Standard | 0.75 | 1.3 | 10 | 9 | 20 | 10 | 7 |
| High carbon | 0.80 | 1.35 | 9.5 | 8.5 | 22 | 10.5 | 7.5 |

## 6. GA-BP Model Validation and Prediction of Building Carbon Emissions

### 6.1. GA-BP Model Validation

After the basic parameters of the genetic algorithm and neural network are set, the GA-BP model is continuously trained using MATLAB software to optimize the parameters as well as the logical structure, to ensure that the training results can fully reflect the actual situation of carbon emission of buildings in Jiangsu Province.

Figures 8 and 9 show that the trained GA-BP model has good performance and is suitable for prediction. In addition, the mean square error of the trained model is 0.034694, which is small, indicating that the model can predict the carbon emissions of buildings in Jiangsu Province more accurately.

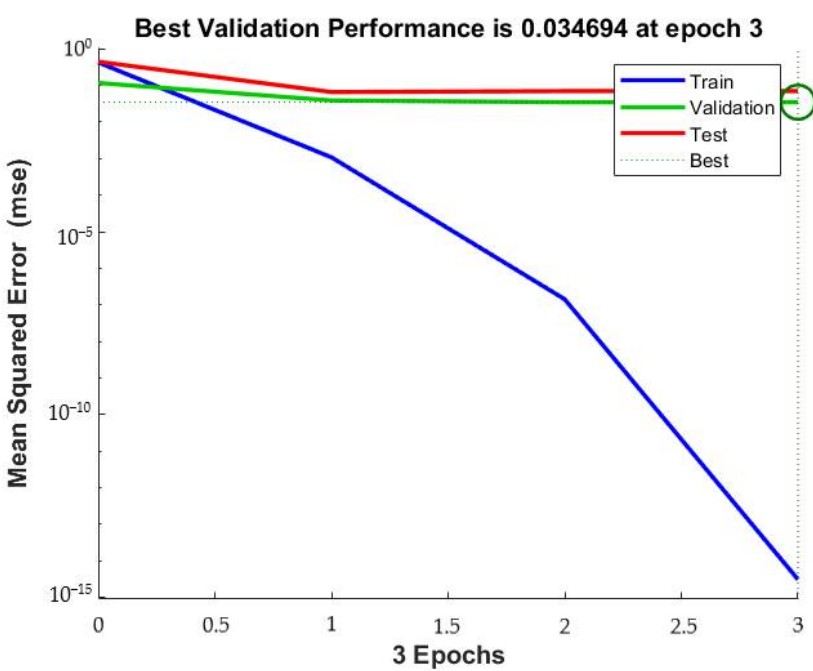

**Figure 8.** Performance analysis of the GA-BP model in training, testing, and validation phases.

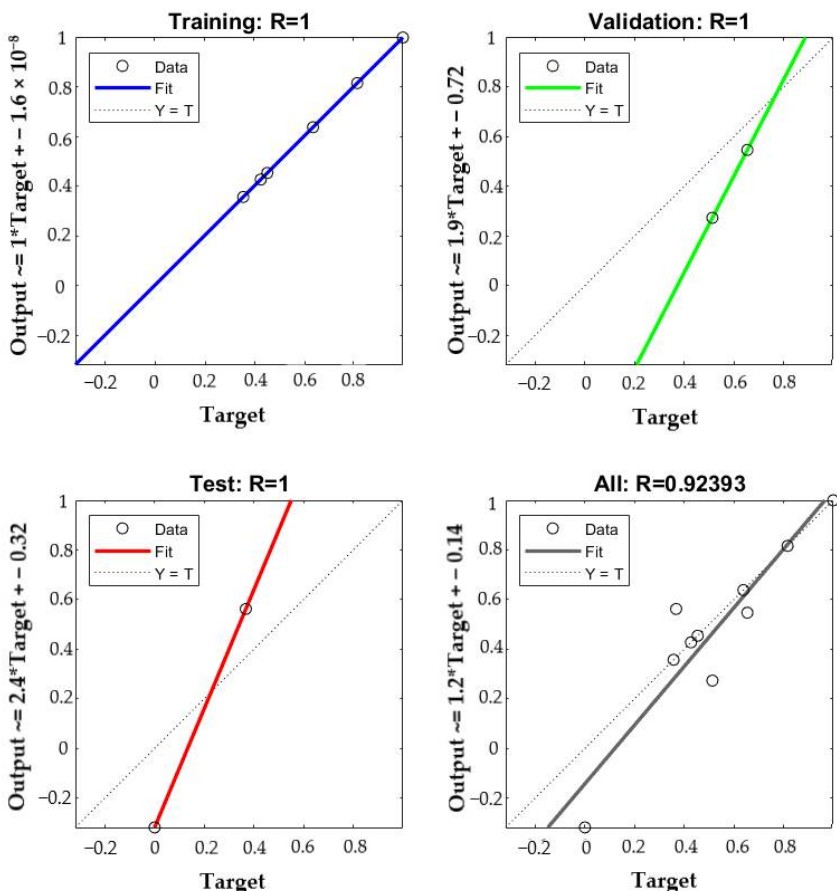

**Figure 9.** Performance comparison of GA-BP model in Jiangsu building carbon emission prediction.

### 6.2. Scenario Prediction Results

As shown in Figure 10, the overall carbon emissions from construction in Jiangsu Province from 2020 to 2024 show a decreasing trend, indicating that the carbon emission reduction work in construction in Jiangsu Province has been practical. By 2024, the carbon

emissions in the low-carbon, baseline, and high-carbon modes will be 437,521,600 tons, 486,433,700 tons, and 531,693,200 tons, respectively, indicating that there is still much room for carbon emission reduction in the construction industry in the province.

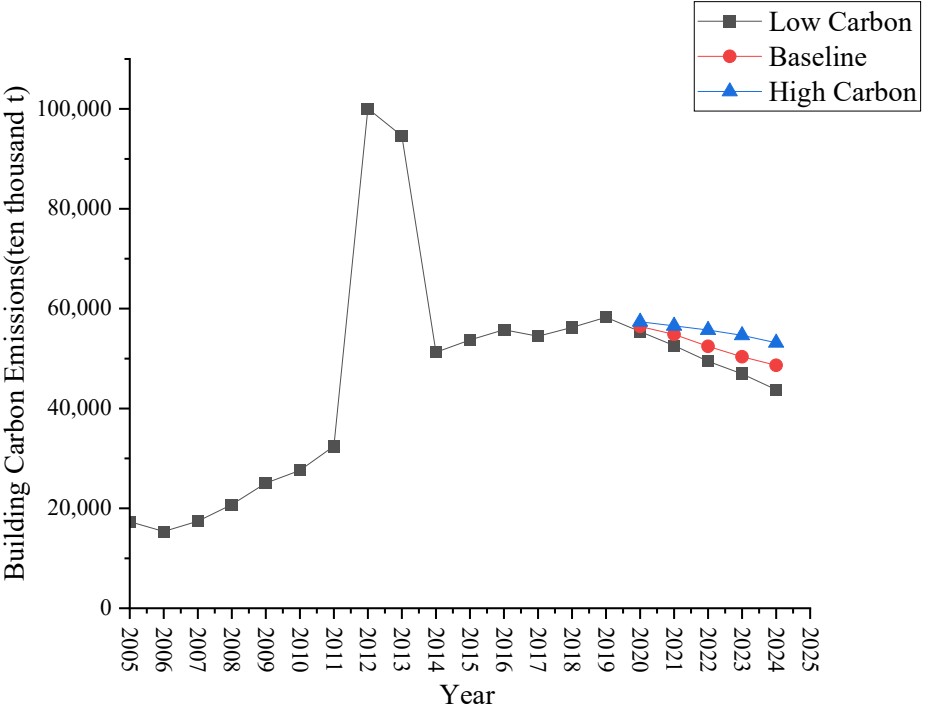

**Figure 10.** Scenario Simulation of Carbon Emissions from Buildings in Jiangsu Province.

## 7. Conclusions

In this paper, we use the STIRPAT model to analyze the factors of carbon emissions from the construction industry in Jiangsu Province and use the GA-BP neural network model to forecast the carbon emissions from construction in Jiangsu Province from 2020 to 2024. The main findings are as follows.

(1) The growth of whole life cycle carbon emissions of buildings in Jiangsu Province from 2005 to 2019 is about 409,476,000 tons, peaking in 2012 and gradually decreasing and leveling off afterward. This shows that the construction industry in Jiangsu Province has the initial carbon peak conditions.

(2) Resident population, urbanization rate, steel production, average road transportation distance, and construction enterprise labor productivity have a catalytic effect on construction carbon emissions; GDP per capita and value-added of the tertiary industry have a suppressive effect. Among them, steel production has a significant catalytic effect on carbon emissions. Therefore, in the process of carbon emission reduction in the construction industry of Jiangsu Province, it is necessary to reduce the use of steel as much as possible and switch to the use of new energy-saving and emission-reducing materials as an alternative.

(3) According to the elasticity coefficient, the absolute value of the coefficient for the resident population is the largest, which indicates that the resident population is the most sensitive factor affecting the change of whole life cycle carbon emission of buildings in Jiangsu province. That is, the change in the number of resident population per unit has the greatest influence on the change of life-cycle carbon emissions of buildings in Jiangsu Province.

(4) The prediction results show that the overall trend of carbon emissions from 2020 to 2024 is decreasing. In addition, under the high carbon scenario and the low carbon scenario, there is a large difference in the carbon emissions of buildings between the

two, indicating that there is still more room for carbon emission reduction in the construction industry in Jiangsu Province.

There are still some limitations that can be improved in future research. Firstly, in this study, only the changes of carbon emission factors of heat and electricity are considered in the whole life cycle carbon emission accounting and forecasting of buildings in Jiangsu Province, and the changes of carbon emission factors of other energy sources can be further considered in future studies to improve the accuracy of accounting and forecasting. Secondly, only some of the major influencing factors are selected for carbon emission prediction study in this study, which will inevitably have an impact on the prediction results. Therefore, all influencing factors can be considered more comprehensively in future studies to reduce the prediction errors. Thirdly, in this study, only three different scenarios of building carbon emissions are predicted, and more scenarios can be set in future studies to predict future building carbon emissions more accurately.

**Author Contributions:** Conceptualization, Methodology, Project administration, original draft, and writing—review & editing Z.H.; data curation, formal analysis, and writing—review & editing, S.Z.; software, visualization, and writing—review & editing, C.Z. All authors have read and agreed to the published version of the manuscript.

**Funding:** This research is supported by the National Key Research and Development Program of China under Grant No. 2021YFB2600600 and Scientific Research and Innovation Plans for Postgraduate of Jiangsu Province under Grant No. KYCX22_3437.

**Institutional Review Board Statement:** Not applicable.

**Informed Consent Statement:** Not applicable.

**Data Availability Statement:** Not applicable.

**Acknowledgments:** The first author thanks Scientific Research and Innovation Plans for Postgraduate of Jiangsu Province (No. KYCX22_3437).

**Conflicts of Interest:** The authors declare no conflict of interest.

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
