# Peer review of "Building Carbon Emission Scenario Prediction Using STIRPAT and GA-BP Neural Network Model"

_sustainability, doi:10.3390/su14159369_

Round 1

Reviewer 1 Report

Please adapt the manuscript to the journal's requirements. There are many errors in the manuscript, e.g.

- you have to remove double citations in the text - please remove the names of the authors of the cited publications in round brackets

- "References" was entered in the wrong place

- the entry of the cited publications in References does not comply with the requirements of the journal

- spaces and periods are missing in many places

- and many other mistakes

Please specify and complete the chapter Conclusions. As should be understood: "In this paper, we use the STIRPAT model to analyze the factors of carbon emissions from the construction industry in Jiangsu Province and use the GA-BP neural network model to forecast the carbon emissions from construction in Jiangsu Province from 2020 is 2024."? Does this mean that the forecasting also covered a period that has partially passed? If so, the entire article will require a thorough revision.

Reviewer 2 Report

The manuscript is well-written. However, the following comments need to be addressed first:

  1. The abstract section needs to be re-written. The problem statement, results and benefits of the developed model should be added to it.
  2. The introduction section needs to be separated from the literature review section. The problem statement and research objectives should be included in the introduction section.
  3. More recent studies need to be added in the literature review section:

a)     Ziyuan, C., Yibo, Y., Simayi, Z., Shengtian, Y., Abulimiti, M., & Yuqing, W. (2022). Carbon emissions index decomposition and carbon emissions prediction in Xinjiang from the perspective of population-related factors, based on the combination of STIRPAT model and neural network. Environmental Science and Pollution Research29(21), 31781-31796.

b)     Al-Sakkaf, A., Mohammed Abdelkader, E., Mahmoud, S., & Bagchi, A. (2021). Studying Energy Performance and Thermal Comfort Conditions in Heritage Buildings: A Case Study of Murabba Palace. Sustainability13(21), 12250.

c)     Zhang, Y., Liu, C., Chen, L., Wang, X., Song, X., & Li, K. (2019). Energy-related CO2 emission peaking target and pathways for China's city: A case study of Baoding City. Journal of Cleaner Production226, 471-481.

  1. It is not clear what are the distinctive features of the proposed optimal field sampling method.
  2. Limitations of previous studies should be added to the manuscript.
  3. A research framework figure and section should be presented to show the steps of the developed model.
  4. Model validation and results need to be collected in a separate section towards the end of the manuscript.
  5. The conclusion section should be strengthened. More insight into the result should be added.
  6. Limitations of the present research study should be added at the end of the conclusion section.

Round 2

Reviewer 1 Report

The revised manuscript may be accept in present form.